# Effects of Tartrazine on Some Sexual Maturation Parameters in Immature Female Wistar Rats

**DOI:** 10.3390/ijerph191610410

**Published:** 2022-08-21

**Authors:** Elisabeth Louise Ndjengue Mindang, Charline Florence Awounfack, Derek Tantoh Ndinteh, Rui W. M. Krause, Dieudonne Njamen

**Affiliations:** 1Department of Animal Biology and Physiology, Faculty of Science, University of Yaounde I, Yaounde P.O. Box 812, Cameroon; 2Department of Chemistry, Faculty of Chemical and Pharmaceutical Sciences, Rhodes University, Makhanda P.O. Box 94, South Africa; 3Department of Psychology, Faculty of Arts, Letters and Social Sciences, University of Yaounde I, Yaounde P.O. Box 7011, Cameroon; 4Department of Chemical Sciences, Faculty of Science, University of Johannesburg, Johannesburg P.O. Box 17011, South Africa

**Keywords:** food additive, tartrazine, rat, early puberty, folliculogenesis, endocrine disruptor

## Abstract

Over the past century, the average age for onset of puberty has declined. Several additives present in our food are thought to contribute significantly to this early puberty which is recognized to also affect people’s health in later life. On this basis, the impact of 40-days unique oral administration of the food dye tartrazine (7.5, 27, and 47 mg/kg BW doses) was evaluated on some sexual maturation parameters on immature female Wistar rats. Vaginal opening was evaluated during the treatment period. At the end of the treatments, animals were sacrificed (estrus phase) and the relative weight of reproductive organs, pituitary gonadotrophin and sexual steroids level, cholesterol level in ovaries and folliculogenesis were evaluated. Compared to the control group, animals receiving tartrazine (47 mg/kg BW) showed significantly high percentage of early vaginal opening from day 45 of age, and an increase in the number of totals, primaries, secondaries, and antral follicles; a significant increase in serum estrogen, LH and in uterine epithelial thickness. Our findings suggest that tartrazine considerably disturbs the normal courses of puberty. These results could validate at least in part the global observations on increasingly precocious puberty in girls feeding increasingly with industrially processed foods.

## 1. Introduction

Puberty is a major developmental event at the end of the juvenile stage, with marked physical and psychological changes, which prepare for adulthood [1,2]. It can be seen as a complex sequence of biological events marked by the reactivation of the hypothalamus-pituitary-gonadal axis after a period of quiescence during childhood; followed by an important increase in sex hormone secretion by the gonads, which leads to a gradual maturation of sexual characteristics which culminate into the attainment of full adult reproductive capacity [1,2,3]. The onset of secondary sexual characteristics, and the pubertal growth spurt, are markers of this developmental process that lead to sexual and reproductive maturity, the development of mental processes and adult identity [4]. Puberty is accompanied by bodily changes, encourages curiosity, promotes interest in sexual activity, increases aggression in adolescents, and can intensify risky behaviors [3]. In the early part of this century, the timing of puberty has received considerable attention because of its associations with health problems such as the increased risk of developing reproductive cancers (breast and prostate cancer), weight gain, obesity and cardiovascular disease later in life. It is also associated with psychosocial problems (depression) and is a risk factor for teenage pregnancy [5,6].

Precocious puberty, or abnormally early sexual development, is defined as the premature onset of pubertal development or secondary sexual characteristics before the age of 8 years in girls and 9 years in boys [4,7]. In girls, it is manifested by features such as advanced breast and ovarian development and rapid bone growth or maturation. It can be attributed to endocrine disorders, with elevated sex hormone secretion [4]. Most recent data consistently indicate that the age of onset of puberty is becoming earlier in Europe and the USA [1] than in Africa [2]. This decrease concerns both the mean age at menarche and the mean age at onset of breast development, which has decreased in all ethnic groups [7,8,9]. These geographical variations indicate an earlier onset in the USA (8.8–10.3 years) and later onset in Africa (10.1–13.2 years) [2]. This may be the result of more stable socio-economic conditions, improved nutrition, and hygiene, as has been observed in industrialized populations. The prevalence of precocious puberty is about 10 times higher in girls than in boys [1,6].

The exact mechanisms underlying the reactivation of the hypothalamus-pituitary-gonadal axis are not fully understood; however, it is assumed that genetic, nutritional, stress-related, and environmental factors influence the onset of puberty [2,6,10]. These include environmental factors such as weight, fetal nutrition, childhood eating habits, physical activity, psychological factors, and exposure to electromagnetic fields and/or endocrine disrupting chemicals [7]. It has been reported that dietary habits appear to significantly influence the mechanism of estrogen metabolism, which is inextricably linked to early puberty [11,12]. Some animal studies have suggested that postnatal overfeeding tends to invariably increase secretion of luteinizing hormone (LH), follicle stimulating hormone (FSH), leptin and insulin levels in pubertal females [13]. Overeating, including excessive consumption of processed foods, is considered the main agent responsible for the secular decline in the pubertal age [13,14]. Today, the growth of consumption of these processed foods is the source of a lively debate about the release of toxins (in concentrations and varieties) into the environment, which damages the endocrine system not only of animals but also that of man [15,16,17]. Thousands of them are widely distributed in food, water, air, and certain industrial products (drugs, cosmetics, and phytosanitary products among other) [15,16,18]. These so-called hormonal (or anti-hormonal) environmental contaminants are designated by the term endocrine disruptors [19].

Many concerns (such as early puberty, declining fertility and cancers) are emerging about the long-term effects on human health following chronic exposure to these substances [16,20,21,22]. Children are more at risk because of intensive use of food additives [16,20,21]. The increased cases of gynecomastia are mostly attributed to food additives (“fast food”, and drinks,) [23]. Unfortunately, few studies on their effects on women’s reproductive function are available. In addition, compared to boys, alterations in the female tract are likely to remain invisible until they reach sexual maturity (puberty).

The reproductive maturation which is a prerequisite of fertility [24] is usually marked in female rats by the vaginal opening [25,26]. The estrous cycle following, is divided in four main phases including proestrus, estrus, metestrus and diestrus characterized by different cell types desquamated from the vaginal epithelium, the presence or absence of leukocytes and mucus in vaginal smears [27,28,29]. Ovulation (sexual receptivity or heat) occurs during the night of the estrus phase after the luteinizing hormone (LH) surge [29,30].

Faced with the hypothesis that food additives contribute to current decline of the age of puberty onset, it becomes urgent not only to identify and quantify each substance that is used as food additives in our diet but also to evaluate the potential endocrine disrupting activity of food additives at concentrations lower than or equal to their threshold of toxicity [31]. In line with this, the present study was designed and carried out to evaluate the potential endocrine disruptor activity on some parameters of sexual maturation in immature female Wistar rats with their hypothalamic-pituitary-ovarian still nonfunctional. In addition, the animal’s diet was soy free, to eliminate any interference with natural phytoestrogens [26,32].

To bring out our contribution, one of the most used dyes (tartrazine) has aroused our interest and justifies its use in this study. Known in some cases as E102, FD and C Yellow 5, C.I 19140, acid Yellow 23, Food yellow 4 or trisodium 1-(4-sulfonatophenyl)-4-sulfonatophenylazo)-5pyrazolonz-3-carboxylate, it is a synthetic lemon-yellow azo dye made by coal tar [33,34], with the chemical formula: 4-5-Dihydro-5-oxo-1-(4-sulfophenyl)4-((4-sulfophenyl) aso) 1H-pyrazole-3 carboxylic acid [35]. In Human, the daily mean exposure of tartrazine is 7.5 mg/kg BW equivalent 47 mg/kg BW in rats [36,37]. Tartrazine is an orange water soluble powder very widespread (drinks, cookies, confectionery, preserves, yogurts, cosmetics, drugs, etc.) used as a dye. It has been shown in previous in vitro studies to be responsible for allergies, tumor diseases, mutagenic and genotoxic effects, and neuro- behavioral disorders (hyperactivity and sleep disturbance in children) [22,38,39,40]. Prolonged usage of tartrazine increases the number of gastric mucosa lymphocytes and eosinophils [41]. Several studies showed that, this dye has adverse effects on male reproduction especially on sperm parameters (negative impact on sperm maturation process and decrease in sperm density, mobility and viability) [33,42]. These effects are accompanied by a significant decrease in serum testosterone concentration [42]. However, the combined treatment of tartrazine and erythrosine mixture in adult male rats impair testicular architecture and function and is accompanied by an increase in a serum hormone (LH, FSH and testosterone) [43]. In female rats, the frequent intake or increased of tartrazine affect thyroid and reproductive hormones (LH, FSH, estrogen, progesterone) and mineral content in tissues; increases the chances of free radical production, leading to the development of oxidative stress in the body [34]. Tartrazine has also been classified as a xenoestrogen [44,45] that can bind to Estrogen Receptor α (ERα) in the Michigan Cancer Foundation-7 (MCF-7) cell line and induce a proliferative effect in breast cancer cells and increase the expression of an estrogen reporter gene [46]. Despite the multiple effects of tartrazine, especially on reproductive hormones [34,42,43,44,45] which are responsible of sex maturation, such as folliculogenesis, ovulation, reproductive behaviors and successful of pregnancy, there is still a lack of available information about its harmful effects on female reproductive function in juvenile. However, no study has evaluated the effects of certain products (tartrazine), considered as potential endocrine disruptors on the parameters of sexual maturation. Therefore, this work aimed at assessing the effects of tartrazine on the physiological parameters allowing the onset of puberty (age of vaginal opening), production of pituitary gonadotropins (FSH and LH), and sex steroids (estradiol and progesterone) to assess their ability to stimulate the hypothalamic-pituitary-ovarian axis; and further, to evaluate their effects on folliculogenesis, on the growth of the reproductive organs (ovaries, vagina, and uterus) in a model of immature female Wistar rats.

## 2. Materials and Methods

### 2.1. Chemicals

Tartrazine (CAS 1934-21-0, Purity ≥ 85%), was purchased from Sigma Aldrich (Munich, Germany). In this study, the doses of 7.5, 27, and 47 mg/kg BW were extrapolated according to the recommendations of daily doses administered by the World Health Organization [47,48].

### 2.2. Animals and Housing

In this case, 21 and 22-day-old immature female Wistar rats (average weight of 30 g) were kept in the animal house with a 12 h of the light-dark cycle. These animals were bred in the laboratory of animal physiology, University of Yaoundé I (Yaoundé, Cameroon) under natural conditions and had free access to diet and drinking water *ad libitum*. Animals housing and experiments were carried out according to the guidelines of the Institutional Ethics Committee of the Cameroon Ministry of Scientific Research and Innovation (Reg. no. FWA-IRD 0001954, 4 September 2006), which has adopted the guidelines established by the European Union on Animal Care CEE Council 86/609).

### 2.3. Dose and Concentation Calculation

In the literature, the human equivalent dose (HED) of tartrazine is 7.5 mg/kg BW per day [36,37]. The animal equivalent dose (AED: 47 mg/kg BW per day) was calculated on the basis of the body surface area, dividing the HED dose (mg/kg BW) by the ratio (km) provided by the literature (AED = HED/km; km = body weight (kg)/body surface area (m^2^). The ratio km used in this work was 0.162 [34,49]. The third dose (27 mg/kg BW per day) used in this work was the mean of the HED and AED.

### 2.4. Experimental Design

The animals (20) were randomly divided into four groups of 5 animals each, a control group that received distilled water and the three test groups which received tartrazine at 7.5, 27 and 47 mg/kg BW. Tartrazine was dissolved in distilled water. The volume of water and substance administered was 10 mL/kg BW. All animals were orally treated by gavage once daily (between 9 and 10 am) for 40 days from the postnatal days 21 to 22. The animals were weighed twice a week and the vaginal opening which is the marker for puberty onset was daily checked until the day it occurred. From the day 36 of treatment (a day when there is vaginal opening in all the animals) until the fortieth day, the animals were sacrificed (in estrus) by decapitation after light anesthesia by diazepam-ketamine i.p. injection (10 and 50 mg/kg BW, respectively). Blood samples were collected for biochemical analysis in dry tubes. The ovaries, uteri, mammary gland and vagina, were dissected and weighed (except the vagina and mammary gland which were immerse immediately in formol). The left ovary and uterus from each animal, as well as the vagina, and mammary glands were fixed in 10% formaldehyde for histological analysis. The right ovary and right uterus were cut, weighed and ground separately with the glass potter’s in sodium phosphate buffer (0.1 M; pH 7.1) to obtain a final homogenate of 20%. After centrifugation at 3000 rpm (Goget Centrifuge, HETTICH, Westphalia, Germany) for 15 min at 5 °C, the collected supernatant was stored at −20 °C for subsequent determination of total uterine and ovarian proteins, and ovarian cholesterol. Blood samples collected in dry tubes were also centrifuged at 3000 rpm at 5 °C for 15 min and the serum obtained was kept at −20 °C until use.

### 2.5. Measurement of Biochemical Parameters

Serum and homogenates of the uterus and ovary were used for biochemical analysis. In the serum, follicle Stimulating Hormone (FSH), (Luteinizing Hormone (LH), Estradiol, and Progesterone were measured in duplicate using the ELISA technique and reagent kits obtained from Cypress Diagnostics (Langdorp, Belgium) according to the manufacturer’s instructions, and precise (intra and inter essay coefficients of variability) with CV ≤ 9.5645% for all the tested samples. Whatever, the total cholesterol in ovaries were measured using reagent kits from Chronolab Systems (Barcelona, Spain).

### 2.6. Histopathological Evaluation

In addition, 5 µm thick sections of paraffin-embedded tissues (uterus, vagina, and ovaries) were prepared and stained with hematoxylin-eosin. The photomicroscopic observation/analysis (uterine and vaginal epithelial thickness, identification of ovarian follicles) was performed using a complete set of Zeiss (Hallbermoos, Germany) equipment (microscope Aioskop 40), the software programs MRGrab 1.0 (Carl Zeis, Hallbermoos, Germany, 2001) and Axio Vision 3.1 (Carl Zeis, Hallbermoos, Germany, 2001) installed in a computer. As concerns folliculogenesis, the tenth section of each ovary was selected. We considered as primary the follicles composed of oocytes surrounded with one layer of cuboidal follicular cells, secondary preantral follicles those with more than one follicular cell layer, and antral follicles those with present antrum of follicular fluid. Ruptured follicles with hypertrophied follicular, cells cavity, and cavity filled with blood were considered as corpora lutea.

### 2.7. Statistical Analysis

Data were expressed as mean ± standard error on the mean (SEM). A two-way ANOVA repeated measures followed by Bonferroni post-hoc tests was used to compare the effect of tartrazine on body weight and the percentage of animals with vaginal opening. The fixed effects or factors were treatment (each individual dose of tartrazine vs. control group), time or periods of analysis, and their interaction. ANOVA one-way followed by Dunnet’s test (when appropriate) was used for the other data with treatment as a fixed effect. All of these tests were performed using GraphPad Prism 5.03 software (La Jolla, CA, USA, 2009). Differences were considered significant at *p* ˂ 0.05.

## 3. Results

### 3.1. Bodyweight of Animals

Figure 1 shows the effect of tartrazine exposure on body weight evolution throughout the period of treatment. Two-way ANOVA indicated a significant time effect (F = 108.3; *p* < 0.0001; df = 13), a non-significant treatment effect (F = 0.2327; *p* = 0.8736; df = 3) and a non-significant interaction effect (F = 0.1026; *p* > 0.9999; df = 39). In addition, Bonferroni correction multiple comparison test indicated that all doses of tartrazine were neither effective (*p* > 0.9999) in increasing, nor reducing body weight suggesting that the significant time effect observed is due to the normal growth of animals over time.

### 3.2. Vaginal Opening

The main endocrine effect on the onset of puberty is summarized in Figure 2. Compared to the control group, the treatment did not induce any significant modification in the mean age of the vaginal opening (Figure 2A) as determined by one-way ANOVA (F = 2.748, *p* = 0.0541). With respect to the percentage of animals with vaginal opening (Figure 2B), the two-way ANOVA showed significant time (F = 38.48; *p* < 0.0001; df = 18) and treatment (F = 10.83; *p* < 0.0001; df = 3) effects. Based on Bonferroni post hoc test, 47 mg/kg BW tartrazine showed a significant difference (*p* < 0.05) in the percentage of animals with vaginal opening as compared with the Control group. This group displayed 100% of vaginal opening vs. 41.66% for control on day 45.

### 3.3. Relative Weight of Ovary and Uterus

The main effects on reproductive organs are presented in Figure 3. The results of the relative uterine weight indicate that there is a statistically significant difference between groups as determined by one-way ANOVA (F = 6.098, *p* = 0.0057). Dunnett Post Hoc multiple comparisons test showed that the difference between tartrazine at the dose of 47 mg/kg BW (Figure 3B) and the control group is statistically significant (*p* < 0.05). Tartrazine increased significantly the relative uterine weight at the dose of 47 mg/kg BW (Figure 3B). Whatever, after 40 days of treatment, tartrazine had no significant effect on the relative weight of the ovaries (Figure 3A) as determined by one-way ANOVA (F = 2.793, *p* = 0.0739).

### 3.4. Epithelial Thickness of the Uterus and Vaginas

The main reproductive effects are summarized in Figure 4. The results of the uterine epithelial thickness indicate that there is a statistically significant difference between groups as determined by one-way ANOVA (F = 6.602, *p* = 0.0041). Dunnett Post Hoc multiple comparisons test showed a significant difference between tartrazine at the dose of 27 (*p* < 0.05) and 47 mg/kg BW (*p* < 0.01) as compared to the control group (Figure 4A). This difference is also confirmed by the microphotographs presented in Figure 4C. However, one-way ANOVA indicated that sacrificed at the estrus phase, the administration of tartrazine had no significant effect (F = 0.6609, *p* = 0.5880) on the vaginal epithelial thickness (Figure 4B).

### 3.5. Mammary Glands

The main effects on mammary glands are summarized in Figure 5. The microphotographs presented showed that, compared to the control group, tartrazine at the dose of 47 mg/kg BW induced eosinophilic secretions in the acinar of the mammary glands.

### 3.6. Ovarian Follicles

The main effects on follicular growth are summarized in Table 1 and Figure 6. Table 1 and Figure 6 show the number of total follicles and different types of follicles, and the microphotographs of ovaries after 40 days of treatment with tartrazine, respectively. One-way ANOVA indicated a significant difference between groups. The difference was reflected on the number of total follicles (F = 8.831, *p* = 0.0011), primary follicles (F = 9.771, *p* = 0.0007), secondary follicles (F = 4.744, *p* = 0.0149), and antral follicles (F = 4.329, *p* = 0.0205). Dunnett Post Hoc multiple comparisons test showed a significant increase in the total number of follicles (*p* < 0.01), primary follicles (*p* < 0.01), secondary follicles (*p* < 0.01), and antral follicles (*p* < 0.05) with tartrazine at a higher dose (47 mg/kg BW) as compared to the control group.

### 3.7. Ovarian Total Cholesterol and Proteins

Figure 7 represents the ovarian total cholesterol and protein after 40 days of treatment. One way ANOVA indicated that ovarian total cholesterol (F = 0.6152, *p* = 0.6151) and protein (F = 0.6086, *p* = 0.6190) were not significantly affected following treatments (Figure 7).

### 3.8. Hormone Levels

The main effects on hormone serum concentrations are summarized in Figure 8. One-way ANOVA indicated a significant difference between groups. The difference was reflected on the LH serum concentration (F = 10.85, *p* = 0.0004), and estradiol serum concentration (F = 130.1, *p <* 0.0001). Dunnett Post Hoc multiple comparisons test showed a significant increase in LH (*p <* 0.001) (Figure 8B) and Estradiol (*p <* 0.001) (Figure 8C) serum concentration at the dose of 47 mg/kg BW as compared to the control group. However, FSH (F = 2.619, *p* = 0.0866) and Progesterone (F = 0.2903, *p* = 0.8318) serum concentrations were not significantly affected by the treatment with tartrazine at all tested doses (Figure 8A,D).

## 4. Discussion

The food additives present in industrial processed food (in the form of preservatives and dyes) are more and more pointed out as being an endocrine disruptor which can be one of the causes of the precocity or the delay of puberty responsible for the current decline of fertility in human [20,23,31]. Several studies have established its effects on male and female reproductive system [33,34,42,43,44,45,46]. This study began with the determination of the age of onset of sexual maturation in experimental animals, which is characterized by vaginal opening [25,26,50,51]. Therefore, this work aimed at assessing the effects of tartrazine on sexual maturation and on folliculogenesis in a model of immature female Wistar rats. Sexual maturation also known as puberty is a crucial stage of development. It requires changes in the sensitivity, activity, and functionality of the hypothalamic-pituitary-gonadal axis which can cause a direct maturation of the genitals [24]. Without these signals, the genitals maintain the appearance they had during childhood, and the reproductive system remains non-functional. In other words, sexual maturation is a prerequisite for fertility [24]. Although there was no effect on the average age of vaginal opening, the results of this study shown that 45 days after their birth, the rats treated with tartrazine (47 mg/kg BW) had (100%) vaginal opening versus 41.66% in animals in the control group of the same age. This result showed that tartrazine advanced puberty as measured by the percentage of rats showing vaginal opening and it is in accordance with Kriszt and colleagues [25] who demonstrated that the administration of xenoestrogen to sexually immature rats is responsible of early puberty. The results showed this advanced puberty is accompanied by a very significant increase in the secretion of estradiol and LH in the group treated with tartrazine at a dose of 47 mg/kg BW. These results corroborate with the observations made in rats by Ramirez and Sawyer [50], which stipulate that the vaginal opening is the initial and external sign of the increase in secretion in estrogens accompanying the beginning of puberty. Tartrazine could have advanced the puberty by activating the release of the reproductive hormones of the hypothalamic-pituitary-gonadal axis. Contrary to the present study, Shakoor and colleagues [34] showed that tartrazine significantly decreases LH, FSH and estrogen levels; and increases progesterone levels after 30 days of treatment at the dose of 9.5 mg/Kg BW to adult female Sprague Dawley rats (6–7 months old). These differences may be due to the differences in species, animal age, and duration of treatment. Literature shows that differences in certain results in studies of the same molecules, substances, plants may arise from differences in the protocols used such as type of studies (in vitro or in vivo); species and age of animal used, duration of treatment, route of administration [52] and the stage of the estrous cycle if the authors used female animals [53]. It is well known that GnRH released in a pulsatile pattern of rhythmic secretory bursts whose amplitude and frequency vary according to cycle stage. The pituitary cells called gonadotropes, responding to GnRH stimulation, synthesize and release LH and FSH, which induce ovarian folliculogenesis, steroidogenesis, ovulation and formation of corpus luteum [24,54]. Furthermore, the increase in serum LH levels after 40 days of treatment observed, testifies the capacity of the rats to ovulate, since the increase in the serum level of gonadotropin to a certain threshold is a prerequisite for ovulation and subsequent promotion of the luteal phase (production of progesterone). According to Marieb and Hoehn [24], the production of estrogens increases with follicular growth and when their size (in this case that of the dominant follicle) reaches a certain threshold, the level of estrogen produced briefly exerts a retro activation on the hypothalamus and the adenohypophysis causing a sudden release of LH to a certain extent and FSH, approximately in the middle of the cycle. This hypothesis confirms the results observed on folliculogenesis (47 mg/kg BW). It increased the number of total follicles (primary, secondary, and antral follicles) and the concentration of FSH (non-significant increase). During puberty, the growth and maturation of the ovaries are mainly attributed to the presence of mature follicles (antral and Graafian follicles [55,56]. Then development and function (size expansion and the number of mature follicles, proliferation of fibrous tissues and antrum) of these ovaries require the presence of estrogens as much as that of pituitary gonadotropins [56]. Meanwhile, the increase of folliculogenesis can lead to the loss of pool of primordial follicles and ovarian follicular reserve is tartrazine is prolonged use from a young age. Literature shows that, when the follicular pool reaches thousand follicles, the ovary cannot maintain the hormonal feedback with the hypothalamus and ovarian ageing also known as menopause is reach [57,58,59]. However, for some authors, the above-mentioned parameters remain insufficient in the evaluation of the impact of chemicals on the development of the female reproductive system. The relative weight of the uterus and/or the size of the uterine and vaginal epithelia are essential parameters [21,60]. In addition to the vaginal opening, reproductive maturation is associated with the activation of the hypothalamic-hypophysis-ovarian axis, which helps to induce secretion of the appropriate quantities of gonadotropins (FSH, LH) and ovarian hormones (estrogens and progesterone) responsible for the development and function of primary estrogen targets (uterus, vagina, and mammary gland) [21,24]. The effects produced by tartrazine at a dose of 47 mg/kg BW confirm this hypothesis. Tartrazine at this dose induced uterine growth (thickness of the uterine epithelium and the relative uterine weight), and eosinophilic secretions in the acinar of the mammary glands. Estrogens have been reported to stimulate proliferation and differentiation of uterine and vaginal epithelial cells, and eosinophilic secretions in the acinar of the mammary glands [61,62]. In an in vitro study, tartrazine has been identified as a new activator of human estrogen receptors (xenoestrogens). Its mechanism of action is through the activation of ERα receptors expressed in various tissues, but mainly in the uterus, ovary, pituitary gland, vas deferens, adipose tissue; and regulates the expression of more than 2800 genes in mammary gland cell lines [44,45]. In addition, it should be interesting to evaluate the effects of tartrazine on the estrous cycle and sexual behavior which are important parameters of sexual maturation.

## 5. Conclusions

In this study, tartrazine at a dose of 47 mg/kg BW (AED corresponding to admissible daily intake in human) increased the cumulative number of rats with vaginal opening, the relative epithelial thickness and the relative weight of the uterus, and the production of hormones of the hypothalamic-pituitary-ovarian axis (LH and Estradiol). Then, on the growth and maturation of the ovarian follicles, the results showed that tartrazine at the dose of 47 mg/kg BW increased the total number of ovarian follicles. Taken all together, these results might justify at least in part the increasingly early onset of puberty observed in our populations and provide a substantial scientific prove confirming tartrazine as an endocrine disruptor.

## Figures and Tables

**Figure 1 ijerph-19-10410-f001:**
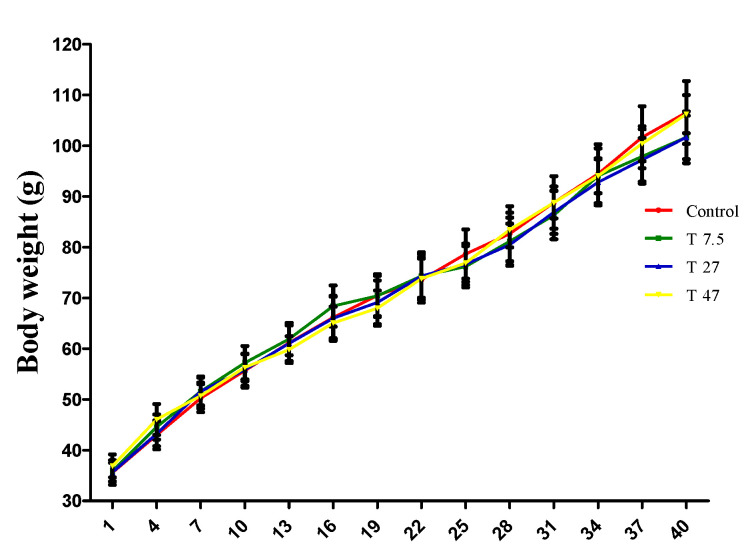
Effects of tartrazine on the bodyweight of immature female Wistar rats during 40 days of treatment. Control = animals receiving vehicle (distilled water, 10 mL/kg BW); T = rats treated with tartrazine at dose of 7.5; 27, and 47 mg/kg BW. Results are shown as a mean ± SEM, *n* = 5.

**Figure 2 ijerph-19-10410-f002:**
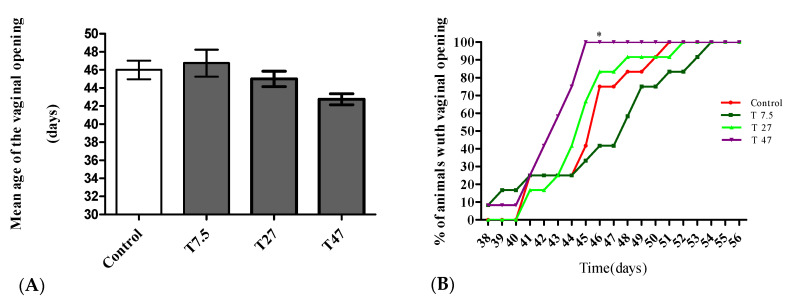
Effects of tartrazine on the mean age of the vaginal opening (**A**), the percentage (%) of rats with a vaginal opening (**B**) during 40 days of treatment. Control = animals receiving vehicle (distilled water, 10 mL/kg BW); T = rats treated with tartrazine at dose of 7.5; 27, and 47 mg/kg BW. Results are shown as a mean ± SEM, *n* = 5. *: *p* < 0.05 in reference to control.

**Figure 3 ijerph-19-10410-f003:**
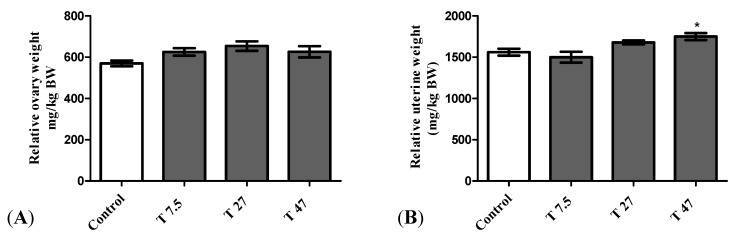
Effect of tartrazine on the relative weight of ovaries (**A**) and uterus (**B**) of immature female Wistar rats after 40 days of treatment. Control = animals receiving vehicle (distilled water, 10 mL/kg BW); T = rats treated with tartrazine at doses of 7.5; 27, and 47 mg/kg BW. Results are shown as a mean ± SEM, *n* = 5. *: *p* < 0.05 in reference to control.

**Figure 4 ijerph-19-10410-f004:**
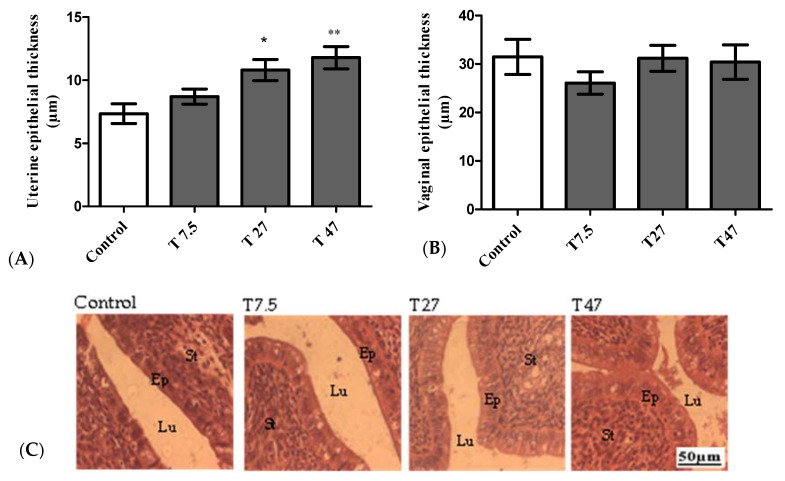
Effect of tartrazine on uterine (**A**) and vagina (**B**) epithelial height as well as microphotographs (25×) of hematoxylin/eosin-stained sections of uteri (**C**) of immature female Wistar rats after 40 days of treatment. Control = animals receiving vehicle (distilled water, 10 mL/kg BW); T = rats treated with tartrazine at doses of 7.5; 27, and 47 mg/kg BW. Results are presented as mean ± SEM; *n* = 5. *: *p* < 0.05; **: *p* < 0.01 compared to the control. Lu = Lumen; Ep = Epithelial cell thickness; St = Stroma.

**Figure 5 ijerph-19-10410-f005:**
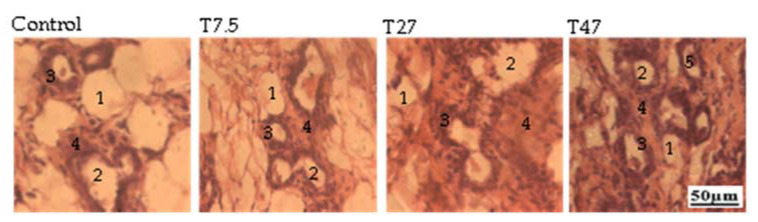
Microphotographs (25×) of hematoxylin/eosin-stained sections of mammary glands of immature female Wistar rats after 40 days of treatment. Control = animals receiving vehicle (distilled water, 10 mL/kg BW); T = rats treated with tartrazine at doses of 7.5; 27, and 47 mg/kg BW. 1 = Adipose tissue; 2 = Lumen of alveoli; 3 = Alveoli epithelium; 4 = Gland parenchyma; 5 = Eosinophilic secretion.

**Figure 6 ijerph-19-10410-f006:**
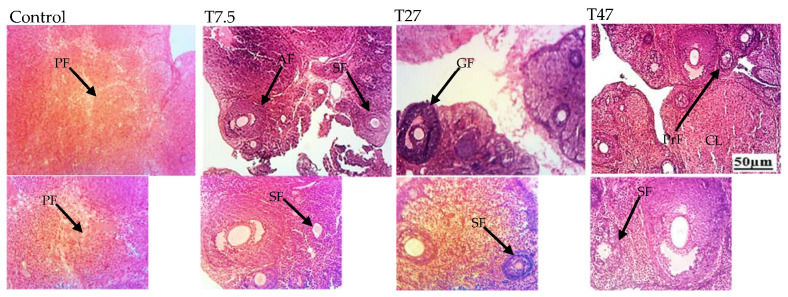
Microphotographs (25× and 200×) of hematoxylin/eosin-stained sections of ovaries of immature female Wistar rats after 40 days of treatment. Control = animals receiving vehicle (distilled water, 10 mL/kg BW); T = rats treated with tartrazine at dose of 7.5; 27, and 47 mg/kg BW. Results are shown as a mean ± SEM, *n* = 5. PF = primordial follicle, PrF = primary follicle, SF = secondary follicle, AF = Antral follicle, GF = Graafian follicle, CL = corpora lutea.

**Figure 7 ijerph-19-10410-f007:**
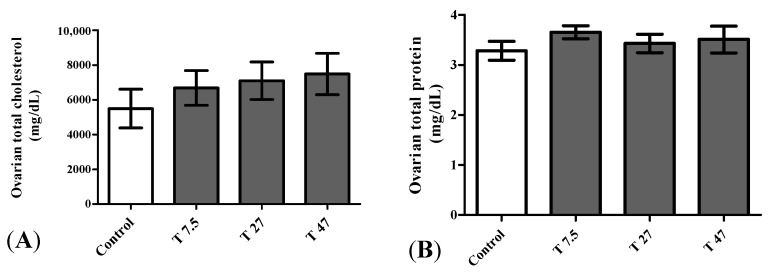
Effect of tartrazine on ovarian total cholesterol (**A**) and ovarian total protein (**B**) of immature female Wistar rats after 40 days of treatment. Control = animals receiving vehicle (distilled water, 10 mL/kg BW); T = rats treated with tartrazine at doses of 7.5; 27, and 47 mg/kg BW. Results are shown as a mean ± SEM; *n* = 5.

**Figure 8 ijerph-19-10410-f008:**
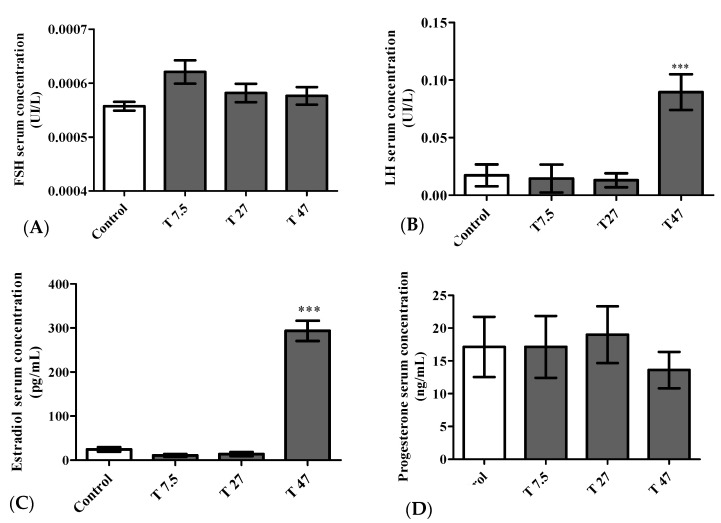
Serum concentration of Follicle Stimulating Hormone (FSH) (**A**), Luteinizing Hormone (LH) (**B**), Estradiol (**C**) and Progesterone (**D**) of immature female Wistar rats after 40 days of treatment. Control = animals receiving vehicle (distilled water, 10 mL/kg BW); T = rats treated with tartrazine at doses of 7.5; 27, and 47 mg/kg BW. Results are shown as a mean ± SEM; *n* = 5. ***: *p* < 0.001 in reference to control.

**Table 1 ijerph-19-10410-t001:** Number of different ovarian follicles and corpora lutea of immature female Wistar rats after 40 days of treatment.

Organs	Control		Tartrazine (mg/kg BW)	
		T 7.5	T 27	T 47
**Total follicles**	92.98 ± 11.90	99.38 ± 11.80	83.49 ± 13.74	145.33 ± 2.37 **
**Primordial follicles**	35.33± 7.83	28.00 ± 5.42	21.33 ± 3.59	38.00 ± 3.83
**Primary follicles**	22.00 ± 2.60	20.33 ± 2.93	18.66 ± 3.00	37.50 ± 2.53 **
**Secondary follicles**	16.50 ± 1.20	30.40 ± 5.76	28.25 ± 8.29	44.25 ± 2.37 **
**Antral follicles**	4.20 ± 0.96	6.25 ± 1.23	4.80 ± 1.59	10.33± 1.42 *
**Graafian follicles**	5.00 ± 1.04	6.80 ± 0.8	4.80 ± 1.01	6.50 ± 0.92
**Atresia follicles**	4.75 ± 0.19	3.80 ± 0.66	3.25 ± 0.58	3.75 ± 0.91
**Corpora lutea**	5.20 ± 1.20	3.80 ± 1.11	2.40 ± 0.60	5.00 ± 0.83

Results are presented as mean ± SEM; (*: *p* < 0.05), (**: *p* < 0.01) in reference to control. Control = animals receiving vehicle (distilled water, 10 mL/kg BW); T = rats treated with tartrazine at doses of 7.5; 27, and 47 mg/kg BW.

## Data Availability

The data presented in this study are not publicly available but available on request from the corresponding author.

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
