# Peer review of "Effects of Tartrazine on Some Sexual Maturation Parameters in Immature Female Wistar Rats"

_ijerph, 2022, doi:10.3390/ijerph191610410_

Round 1

Reviewer 1 Report

attached comments

Author Response

Thank you so much for all the comments which helped us to improve the paper. Please find below the answers.

Experimental design
a.- What was the rationale to select the dose used in the rats? Please discuss in terms of the possible effects and the concentration that the substance could reach in the organs. That is to present the pharmacological reason to select the substance. Why do the authors name pharmacological only the highest dose used. That's means that T is not accumulated? Or
they are metabolized? All these questions permit the authors to project its study and propose what happens in the human with the dose used.

Thank you for bringing up our attention to the term pharmacological dose. In fact, we wanted to talk about the admissible daily intake (ADI) of tartrazine authorized by the Food and drugs administration (FDA, 2005). The term has been changed in the manuscript. Attached is the answer in lines 430 to 431 of the new manuscript.

 Concerning the doses used, in the literature, the human equivalent dose (HED) of tartrazine is 7.5 mg/kg BW per day [JECFA, 1995; Tanaka, 2006]. The animal equivalent dose (AED: 47 mg/kg BW per day) was calculated on the basis of the body surface area, dividing the HED dose (mg/kg BW) by the ratio (km) provided by the literature (AED= HED/km; km= body weight (kg)/body surface area (m2). The ratio km used in this work was 0.162 [FDA, 2005; Shakoor et al., 2022]. The third dose (27 mg/kg BW per day) used in this work was the mean of the HED and AED. Attached is the answer from Lines 154 to 159 of the new manuscript. 

b. Justify or discuss the doses used with the concentrations found of tartrazine in foods and beverage to justify why the only effect found in the paper are the ones that the authors described as pharmacological doses

The choice of the dose used has been made on the basis of the Admissible Daily intake in humans (ADI=7.5 mg/kg BW). Attached is the answer from Lines 154 to 159 of the new manuscript.

According to the observed effects, Tartrazine did not only have effects at the dose of 47 mg/kg BW. We also observed the increase of the epithelium thickness at the dose of 27 mg/kg BW hat could lead later to myomas. Attached is the answer in Figures 4A and 4C of the new manuscript.

At 7.5 mg/kg BW, tartrazine has been responsible for an increase in FSH concentration. Attached is the answer in Figure 8A of the new manuscript.

WE insisted on the dose of 47 mg/kg BW because this dose affected almost all the parameters (increase in the percentage of animals with vaginal opening, the increase in the epithelium uterine thickness, eosinophilic secretion in the acinar of mammary glands, increase in the number of follicles, increase in LH, and estradiol secretion. Attached is the answer in Figures 2B, 4A, 4C, 5, 6, 8B, 8C, and Table 1 of the new manuscript.

Line 130: The authors said that the rats were euthanized in estrus. Please explain what does it means estrus in terms of reproductive physiology. It is very important to correlate the stage of the estrous cycle with the stage of follicular development. Did the authors daily checked for the estrous cycle?. If you did, please present the data to understand if Tartrazine , modified the ovulatory cycling activity and the impact to the main objective of the research. Please revise the following papers: 

The explanation of the estrous cycle has been added to the manuscript. In female rats, reproductive maturation is usually marked by the vaginal opening [Tassinari et al., 2022; Kriszt et al., 2015]. The estrous cycle following, is divided in four main phases including proestrus, estrus, metestrus, and diestrus characterized by different cell types desquamated from the vaginal epithelium, the presence or absence of leukocytes and mucus in vaginal smears [Hubscher and Brooks, 2005; Freeman, 1994; Paccola et al., 2013]. Ovulation (sexual receptivity or heat) occurs during the night of the estrus phase after the luteinizing hormone (LH) surge [Johnson, 2007]. Attached is the answer from lines 87 to 92 of the new manuscript.

The estrous cycle has been checked daily, but we could not evaluate the effect of tartrazine since some of our animals had vaginal opening the week of the sacrifice and we need at least 3 to 4 cycles for this evaluation. Although we did another study where we evaluated the effects of tartrazine on sexual behavior, for this study we evaluated as well the estrous cycle and used for it adult rats. We are still analyzing the result for the publication. Attached are the results of the percentage of vaginal opening in Figure 2B of the new manuscript.

Line 141: Hormones determination. Please add the intra and inter variability of the test used and sensitivity of each assay.

Thank you for the question, we apologize for it. I run this experiment last year when I was in Cameroon, but now I am actually in South Africa to continue the research. I ask them from home, but they could not find the reagent kit and protocol and I could not also find it online.

Results: All data show that estradiol could be the most important change in the T-treated rats. Does estradiol stimulated ovulation?, Did you get some data supporting this suggestion?. Is important to discuss the changes in the hormones data presented in the paper.

The increased estradiol concentration is observed with tartrazine at the dose of 47 mg/kg BW. Although ovulation is not induced by estradiol but by LH. Estradiol is involved in follicular growth. The changes in hormone concentration have been added to the discussion Attached is the answer from lines 370 to 402.

Discussion

The increased data in estradiol modified the ovulation?. Did you follow the estrous cycling activity?. There is no increase in Progesterone in the stimulated rats, normally progesterone is a good marker for the presence of corpus luteum. Is interesting to include a paragraph discussing the changes or no changes in plasma steroids and ovary function. Probably, as results show, there are more follicles. Please give an explanation to this part of the work and also try to include in some model showing the role of the endocrine disruptor in the ovary.

The increased estradiol concentration is observed with tartrazine at the dose of 47 mg/kg BW. Although ovulation is not induced by estradiol but by LH.  Estradiol is also an important hormone implicated in folliculogenesis.

We did not observe an increased number of corpus luteum, this can justify the result obtained in progesterone concentration. Attached is the answer in Table 1 of the new manuscript.

The estrous cycle has been checked daily, but we could not evaluate the effect of tartrazine since some of our animals had vaginal opening the week of the sacrifice and we need at least 3 to 4 cycles for this evaluation. Although we did another study where we evaluated the effects of tartrazine on sexual behavior, for this study we evaluated as well the estrous cycle and used for it adult rats. We are still analyzing the result for the publication. Attached are the results of the percentage of vaginal opening in Figure 2B of the new manuscript.

The changes in hormone concentration have been added to the discussion Attached is the answer from lines 370 to 402.

The potential mechanism of action of oendocrine disruptors on the ovary has been added in the discussion of the new manuscript. Literature shows that folliculogenesis can be responsible of early menopause. It has been shown that when the follicular pool reaches  thousand follicles, the ovary cannot maintain the hormonal feedback with the hypothalamus and ovarian ageing also known as menopause is reach (te Velde, 1998; Wilkosz et al., 2014; Cruz et al., 2017). this can give a potential mecanism of action of substances inducing folliculogenis on health. Attached is the answer from line 406 to 408 of the new manuscript.

Reviewer 2 Report

Comments to the Author

Comments:
Manuscript Number: ijerph-1779790, Title: Effects of tartrazine on some sexual maturation parameters in immature female Wistar rats

 General comments

 The study presents original information on the effects of tartrazine on puberty in female rats. Although the work is original and interesting, it needs future revisions, including a more concrete hypothesis regarding the possible effects of tartrazine, weaknesses in the statistical analysis and in the presentation of results, as well as a poor discussion and conclusions, with some contrary statements to some of the results obtained. It is suggested that the authors review a series of specific comments, which I hope will be helpful to the authors:

 Specific comments

 Abrstact

 L26: our by “Our”

 L27-29: Do the authors not consider this statement to be highly speculative? Well, although the results of this study may be in agreement with the type of diet, future studies are needed to confirm what happens in humans. Why partially explain?

 Keywords

 Avoid repeating words that are already in the title.

 Introduction

 In the introduction, virtually no information on the effects of tartrazine on reproduction is presented from previous reports. How do the authors arrive at the hypothesis?

More specific studies on the action of tartrazine on reproduction need to be presented.

Although there are few papers on tartrazine and reproduction and although most are in males, these should also be discussed in the introduction. However, a recent study was published on female rats.

 For example, authors are required to review the following papers:

 Shakoor et al. 2022. Impact of tartrazine and curcumin on mineral status, and thyroid and reproductive hormones disruption in vivo. International Food Research Journal 29(1), pp. 186-199.

 Wopara et al. 2021. Synthetic food dyes cause testicular damage via up-regulation of pro-inflammatory cytokines and down-regulation of fsh-r and tesk-1 gene expression. Jornal Brasileiro de Reproducao Assistida 25(3), pp. 341-348.

 Boussada et al., Assessment of a sub-chronic consumption of tartrazine (E102) on sperm and oxidative stress features in Wistar rat. International Food Research Journal 24(4), pp. 1473-1481.

 Mehedi et al., 2009. Reproductive toxicology of tartrazine (FD and C Yellow No. 5) in Swiss albino mice. American Journal of Pharmacology and Toxicology 4(4), pp. 130-135.

 Materials and methods

 Describe in more detail how the drug was administered orally to the animals.

 In relation to hormones and other biochemical parameters, report intra- and inter-assay coefficients of variation, sensitivity limits.

 There is a lack of explanation and clarification of the statistical analyses, which is a great weakness of this study.

 For example, do all the data have a normal distribution? With what tests and how did you evaluate the normality of the variable and of the residuals?

 L162-164: Percentage of vaginal opening or percentage of animals with vaginal opening?

 L163: If it is a two-way analysis, it is necessary to clarify which are the two factors or fixed effects. Did they analyze interaction between factors?

 Data that was obtained and recorded repeatedly over time, such as body weight, how were they statistically evaluated? In the statistical analysis section it was not mentioned how these data were evaluated with repeated measurements. What fixed effects and random effects were included in the analysis? Why did they not include the interaction between the fixed effects?

 All these elements are extremely important when analyzing the results and being able to make inferences about them. I suggest the authors consult a statistician so that they can shed light on this type of analysis.

 On the other hand, in addition to the statistical differences, "Effect sizes" should be included.

Results

 VERY IMPORTANT: For all results (3.1, 3.2, 3.3, 3.4, 3.5, 3.6) the first thing that should be shown or presented is the main effect(s) of the ANOVA. For example, was there a main effect of the treatment?

In this sense, the p value of the main effect of the ANOVA must first be shown for each of the variables analyzed.

Only if the main effect of the ANOVA is significant (which value must be reported), only then can differences between groups be sought.

3.1 L167-174: In the statistical analysis, it was not mentioned how body weight was evaluated, nor its fixed effects, nor if interactions were evaluated and how the random effect was included. Please clarify this analysis in more detail.

 L176-177: Was there a main effect of the treatment?

 L177-179: Does this variable have a normal distribution? Half of the times the value 0 is repeated!!!! Check and see how is the best way to analyze this variable. What is the p value of the main effects? Was interaction analyzed? If so, how was the p value of the interaction?

 L195: Figure 3C or 4C?

 L196: What does "a slight increase" mean? It was measured? How was it analyzed? Were there significant differences or not?

 Figure 2B: Check the title of the legend on the ordinate axis.

What does "TEMOIN" mean?

 236: 3.5 or 3.6? Check!!!!

Discussion and conclusion

 The discussion should be redrafted after performing a new statistical analysis of the data. Anyway, I send some general comments.

 L264-279: This paragraph does not discuss any results obtained from the work, so I suggest removing that paragraph or locating it elsewhere in the manuscript. Remember that the discussion is a section of the manuscript intended to precisely discuss the results obtained.

 Everything that is mentioned here can go in the introduction, and even what was done is repeated even until the objective. Why do the authors consider repeating the objective in the discussion section? Avoid repeating information unnecessarily.

 L287-289: In the previous sentences the effect of tartrazine is spoken of, while here it is spoken of puberty, in what sense do they corroborate? and what corroborates?

 L297-300, 308-313: Although the statistics must be reviewed again, as the authors themselves show in their results (Figure 6 A) there were no differences in FSH levels at a dose of 47 mg/Kg. In this sense, how do you explain this result?

 L319-322: This should have been mentioned in the introduction.

 Conclusions

 L325-327: Was there really an effect on organ weight and FSH levels? This contradicts the results presented.

Author Response

Thank you so much for the comments which helped us to improve the manuscript

Responses for the comments.

Abstract

 L26: our by “Our”: It has been corrected in the manuscript. Attached is the answer in line 26 of the new manuscript.

L27-29: Do the authors not consider this statement to be highly speculative? Well, although the results of this study may be in agreement with the type of diet, future studies are needed to confirm what happens in humans. Why partially explain?

We added '' could'' in the abstract of the new manuscript to agree that a study on humans is needed to confirm the results. Attached is the answer on line 27 of the new manuscript.

The results are partially explained because it is not only tartrazine that can be responsible for early puberty. Some endocrine disruptors such as Bisphenol A  (Ricard, 2011; Tassinari et al., 2021), some fatty food with increased in the body weight and the exposition to some movies, also induce precocious puberty

Keywords

 Avoid repeating words that are already in the title: We removed immature Wistar rat and we added food additive and Rat. Attached is the answer on line  30 of the new manuscript.

Introduction

 In the introduction, virtually no information on the effects of tartrazine on reproduction is presented from previous reports. How do the authors arrive at the hypothesis?

Thank you for the articles showing the effects of tartrazine on male and female reproduction. we used them to improve the introduction as well as the manuscript. The information on the impact of tartrazine on reproduction has been added to the manuscript. Several studies showed that has adverse effects on sperm parameters (negative impact on sperm maturation process and decrease in sperm density, mobility, and viability) [Boussada et al., 2017; Wopara et al., 2021]. The effects observed on sperm parameters are accompanied by a significant decrease in serum testosterone concentration [Boussada et al., 2017]. However, the combined treatment of tartrazine and erythrosine mixture in adult male rats impair testicular architecture and function and is accompanied by an increase in a serum hormone (LH, FSH and testosterone) [Mehedi et al., 2009]. In female rats, the frequent intake or increased of tartrazine affect thyroid and reproductive hormones (LH, FSH, estrogen, progesterone) and mineral content in tissues; increases the chances of free radical production, leading to the development of oxidative stress in the body [Shakoor et al., 2022]. Attached is the answer from line 114 to 123 of the new manuscript.

Materials and methods

 Describe in more detail how the drug was administered orally to the animals.

The details on the administration of the drug have been added to the manuscript. The animals were orally treated by gavage once daily (between 9 and 10 am) for 40 days from the postnatal days 21 to 22. Attached is the answer in line 165 of the new manuscript.

 In relation to hormones and other biochemical parameters, report intra- and inter-assay coefficients of variation, sensitivity limits.

We apologize for the information on biochemical analysis. The study has been done one year ago when I was in Cameroon. I am actually in South Africa to continue with the project and I did not travel with the protocol. I asked them from home but we could not find them.

 There is a lack of explanation and clarification of the statistical analyses, which is a great weakness of this study.

 For example, do all the data have a normal distribution? With what tests and how did you evaluate the normality of the variable and of the residuals? 

ANOVA is a normally and independently distributed test with equal variances among groups. However, real data are often not normally distributed and variances are not always equal.

 L162-164: Percentage of vaginal opening or percentage of animals with vaginal opening?

The sentence has been corrected in the new manuscript. We wanted to speak about the percentage of animals with vaginal opening. Attached is the answer on line 204 of the new manuscript.

 L163: If it is a two-way analysis, it is necessary to clarify which are the two factors or fixed effects. Did they analyze interaction between factors?

Yes, Two-way ANOVA analyzes interaction between factors (two factors). The percentage of animals with vaginal opening and the body weight were analyzed with two-way ANOVA. The two factors for the percentage of animals with vaginal opening were time (age of animals)  and the percentage of animals with vaginal opening. The two factors of the body weight were time (age of animal) and body weight. attached is the answer from lines 203 to 207.

 Data that was obtained and recorded repeatedly over time, such as body weight, how were they statistically evaluated? In the statistical analysis section it was not mentioned how these data were evaluated with repeated measurements. What fixed effects and random effects were included in the analysis? Why did they not include the interaction between the fixed effects?

The statistical method used for the analysis of body weight has been added to the manuscript. There was any fixed effect. the factors were different when recording then (age of animal, percentage of animal with vaginal opening, and body weight). The percentage of animals with vaginal opening and the body weight were analyzed with two-way ANOVA. The two factors for the percentage of animals with vaginal opening were time (age of animals)  and the percentage of animals with vaginal opening. The two factors of the body weight were time (age of animal) and body weight. attached is the answer from lines 203 to 207.

 All these elements are extremely important when analyzing the results and being able to make inferences about them. I suggest the authors consult a statistician so that they can shed light on this type of analysis.

 On the other hand, in addition to the statistical differences, "Effect sizes" should be included.

Results

 VERY IMPORTANT: For all results (3.1, 3.2, 3.3, 3.4, 3.5, 3.6) the first thing that should be shown or presented is the main effect(s) of the ANOVA. For example, was there a main effect of the treatment? 

For all the results, we added the main effects of the treatment. Attached are the answers in the new manuscript: 3.1: Lines 212 to 213: The main effect on the body weight analyzed with ANOVA two-way followed by Bonferroni’s test is shown in Figure 1.; 3.2: Line 221:The main endocrine effect on onset puberty is summarized in Figures 2.; 3.3: Line 235: The main effects on ovaries and vagina weight are represented in Figure 3.; 3.4: Line 244 to 246: The main reproductive effects are summarized in Figure 4. The calculated difference between epithelial thickness of uterus and vagina was analyzed with the ANOVA one-way followed by Dunnett’s test.; 3.5: Line 266:The main effects on mammary glands are summarized in Figure 5.; 3.6: Line 282: The main effects on follicular growth are summarized in Table 1 and Figure 6.; 3.7: Line 311:The main effects on ovarian cholesterol and protein presented in Figure 7.; 3.8: Line The main effects on hormones serum concentration are summarized in Figure 8. 

In this sense, the p value of the main effect of the ANOVA must first be shown for each of the variables analyzed.

The P value of the main effect of the ANOVA has been added in the new manuscripts. 3.2: Line 226 to 227; 3.3: Line 239 to 240; 3.4: Line 248 to 249; 3.6: Line 288 to 289; 3.7: Line 314 to 315; 3.8: Line 323 to 324.

Only if the main effect of the ANOVA is significant (which value must be reported), only then can differences between groups be sought.

3.1 L167-174: In the statistical analysis, it was not mentioned how body weight was evaluated, nor its fixed effects, nor if interactions were evaluated and how the random effect was included. Please clarify this analysis in more detail.

The statistical method used for the analysis of body weight has been added to the manuscript. There was any fixed effect. the factors were different when recording then (age of animal and body weight). The body weight was analyzed with two-way ANOVA. The two factors the body weight were time (age of animal) and body weight. attached is the answer from lines 203 to 207.

 L176-177: Was there a main effect of the treatment?

For all the results, we added the main effects of the treatment. Attached are the answers in the new manuscript: 3.1: Lines 212 to 213: The main effect on the body weight analyzed with ANOVA two-way followed by Bonferroni’s test is shown in Figure 1.; 3.2: Line 221:The main endocrine effect on onset puberty is summarized in Figures 2.; 3.3: Line 235: The main effects on ovaries and vagina weight are represented in Figure 3.; 3.4: Line 244 to 246: The main reproductive effects are summarized in Figure 4. The calculated difference between epithelial thickness of uterus and vagina was analyzed with the ANOVA one-way followed by Dunnett’s test.; 3.5: Line 266:The main effects on mammary glands are summarized in Figure 5.; 3.6: Line 282: The main effects on follicular growth are summarized in Table 1 and Figure 6.; 3.7: Line 311:The main effects on ovarian cholesterol and protein presented in Figure 7.; 3.8: Line The main effects on hormones serum concentration are summarized in Figure 8. 

 L177-179: Does this variable have a normal distribution? Half of the times the value 0 is repeated!!!! Check and see how is the best way to analyze this variable. What is the p value of the main effects? Was interaction analyzed? If so, how was the p value of the interaction? 

The variable has been analyzed one more and we saw this mistake. The repeated value 0 has been removed. Attached is the answer in Figure 2B of the new manuscript. After analysis, the p value of the main effect was p ˂ 0.05 showing that there was an increase in the percentage of animal with the vaginal opening at the dose of 47 mg/kg BW compared to the control group.

 L195: Figure 3C or 4C? 

It has been corrected in the new manuscript. We replace 3C by 4C. Attached is the answer on Line 248 of the new manuscript.

 L196: What does "a slight increase" mean? It was measured? How was it analyzed? Were there significant differences or not?

Thank you so much for the remark. We removed it in the manuscript. whatever, this result (Figure 4C)was supporting the result on epithelium thickness (Figure 4A). The epithelium thickness has been measured and presented in Figure 4A. Tartrazine increased this parameter at the doses of 27 and 47 mg/kg BW. Attached is the answer from lines 246 to 249 of the new manuscript.

 Figure 2B: Check the title of the legend on the ordinate axis.

What does "TEMOIN" mean?

The title has been checked and corrected in the new manuscript. We wanted to write ''Control''. Attached is the answer in Figure 2B of the new manuscript.

 236: 3.5 or 3.6? Check!!!!

It has been corrected in the new manuscript. We added the result of mammary glands presented in the discussion. So 3.5 became 3.7. Attached is the answer in line 310 of the new manuscript.

Discussion and conclusion

 The discussion should be redrafted after performing a new statistical analysis of the data. Anyway, I send some general comments. 

The discussion has been redrafted. Attached is the answer from lines 369 to 456 of the new manuscript.

 L264-279: This paragraph does not discuss any results obtained from the work, so I suggest removing that paragraph or locating it elsewhere in the manuscript. Remember that the discussion is a section of the manuscript intended to precisely discuss the results obtained.

 Everything that is mentioned here can go in the introduction, and even what was done is repeated even until the objective. Why do the authors consider repeating the objective in the discussion section? Avoid repeating information unnecessarily.

We delated a big part of this paragraph and added it in the introduction, we only kept what we wanted to use to discuss the results. Attached is the answer from lines 351 to 364 of the new manuscript. 

 L287-289: In the previous sentences the effect of tartrazine is spoken of, while here it is spoken of puberty, in what sense do they corroborate? and what corroborates?

Puberty (also known as sexual maturation) presented here was discussing the early puberty obtained with tartrazine at the dose of 47 mg/kg BW. All the results in the treated groups were compared to the control group.

 L297-300, 308-313: Although the statistics must be reviewed again, as the authors themselves show in their results (Figure 6 A) there were no differences in FSH levels at a dose of 47 mg/Kg. In this sense, how do you explain this result?

The animals were sacrificed during the estrous stage. The animal was not sacrificed on the day of the vaginal opening. We took into consideration the age of puberty for the sacrifice because we wanted all the animals to be sexually mature first.

 L319-322: This should have been mentioned in the introduction.

The xenoestrogenicity of tartrazine has been added in the introduction. Attached is the answer from lines 123 to 126 of the new manuscript.

 Conclusions

 L325-327: Was there really an effect on organ weight and FSH levels? This contradicts the results presented. 

The FSH level increased with tartrazine only at the dose of 7.5 mg/kg BW and there were no changes in organ weight nor epithelium thickness with tartrazine at this dose. Attached is the answer in Figure 3 and Figure 8A.

Reviewer 3 Report

Journal: IJERPH (ISSN 1660-4601)

Manuscript ID: ijerph-1779790

Type:Article

Title:Effects of tartrazine on some sexual maturation parameters in immature female Wistar rats

Authors:Elisabeth Louise Ndjengue Mindang * , Charline Florence Awounfack , Derek Tantoh Ndinteh , Rui W. M. Krause * , Dieudonne Njamen *

Section:Public Health Statistics and Risk Assessment

Special Issue:Challenges and Emerging Approaches in Environmental Exposure and Human Health Risk Assessment

Revision:

The Article entitled: "Effects of tartrazine on some sexual maturation parameters in immature female Wistar rats" investigated the potential endocrine disruptor activity of tartrazine on some parameters of sexual parameters in immature female Wistar rats with their hypothalamic-pituitary-ovarian still non functional. This is of great interest because the tartrazine is used as a dye and it’s very widespread (drinks, cookies, confectionery, preserves, yogurts, cosmetics, drugs, etc.). The article is very interesting  given the increase in the use of food additives in industrial processed food and given that this additives are more and more pointed out as being an endocrine disruptor. I think that the manuscript is suitable for its publication in IJERPH (ISSN 1660-4601), special Issue Challenges and Emerging Approaches in Environmental Exposure and Human Health Risk Assessment, after minor revisions:

In the Introduction section

lines 93-94: the sentence “In line with this, the present study was designed and carried out to evaluate 93 the potential endocrine disruptor activity on some parameters of sexual.......” is repeated 2 times in a row. Delete or rewrite the second.

line 97: “Related to this”....line 99: “To contribute to this”......line 93: “In line with this”....... they are too close together, similar and repetitive; the authors should change them.

In the Materials and Methods section

lines 141-147: subparagraph “2.4 Measurement of Biochemical Parameters” describe how you got homogenates of the uterus and ovary

Author Response

Thank you so much for the comments, they helped us to improve the manuscript.

Answers to the comments

In the Introduction section

lines 93-94: the sentence “In line with this, the present study was designed and carried out to evaluate is repeated 2 times in a row: We deleted one sentence. Attached is the answer in line 97 of the new manuscript.

line 97: “Related to this”....line 99: “To contribute to this”......line 93: “In line with this”....... they are too close together, similar and repetitive; the authors should change them: We changed the transitions. Related to this by ''in addition''; To contribute to this by ''To bring out our contribution''. Attached is the answer in line  Line 99 and 102 respectively of the new manuscript.

In the Materials and Methods section

lines 141-147: subparagraph “2.4 Measurement of Biochemical Parameters” describe how you got homogenates of the uterus and ovary:

The preparation of the homogenates of the uterus and ovary is described in the experimental design. Attached is the answer from lines 175 to 180 of the new manuscript.

Round 2

Reviewer 1 Report

No additional comments

Author Response

Thank you so much for your first review which helped us to improve the manuscript

Reviewer 2 Report

A couple of comments:

a) I reviewed the new version of the manuscript, and although the authors mention in the results for example: "The main effect on the body weight analyzed with ANOVA two-way followed by 212

Bonferroni's test is shown in Figure 1." Actually the authors do not present the main effect of the ANOVA, what is the exact p-value of the fixed effect of the treatment? That value was not presented. This for all the results. On the other hand, in In relation to the question that I had asked in the previous review of whether the fixed effect of time and the interaction between treatment and time were included in the statistical analysis?

This was not clarified in Materials and methods nor do they present such results. I reiterate that these elements are basic in data analysis, and therefore I had suggested that you consult a statistician.

b) At each beginning of each result, it begins with some phrases as shown for example in L212-213, 221-225, etc... but the statistical analysis should not be repeated in results, therefore that should be in Materials and Methods in statistical analysis. Furthermore, I repeat again, that the main effects and the other fixed effects and interactions were not presented.

It will be up to the editors if the article can continue with these faults.

Author Response

Thank you so much for the additional comments which helped us to improve the manuscript.

a) I reviewed the new version of the manuscript, and although the authors mention in the results for example: "The main effect on the body weight analyzed with ANOVA two-way followed by 212

Bonferroni's test is shown in Figure 1." Actually the authors do not present the main effect of the ANOVA, what is the exact p-value of the fixed effect of the treatment? That value was not presented. This for all the results. On the other hand, in In relation to the question that I had asked in the previous review of whether the fixed effect of time and the interaction between treatment and time were included in the statistical analysis?

This was not clarified in Materials and methods nor do they present such results. I reiterate that these elements are basic in data analysis, and therefore I had suggested that you consult a statistician.

We consulted a statistician. The main effect of ANOVA has been added for all the results. The P value of the fixed effect and the interaction between treatment and time have been included in the manuscript. Attached is the answer in the new manuscript:

Lines 213 to 220

3.1. Bodyweight of animals

Figure 1 shows the effect of tartrazine exposure on body weight evolution throughout the period of treatment. Two-way ANOVA indicated a significant time effect (F = 108.3; p < 0.0001; df = 13), a non-significant treatment effect (F = 0.2327; p = 0.8736; df = 3) and a non-significant interaction effect (F=0.1026; p > 0.9999; df=39). In addition, Bonferroni correction multiple comparison test indicated that all doses of tartrazine were neither effective (p > 0.9999) in increasing, nor reducing body weight suggesting that the significant time effect observed is due to the normal growth of animals over time.

Lines 225 to 234

3.2. Vaginal opening

The main endocrine effect on the onset of puberty is summarized in Figure 2. Compared to the control group, the treatment did not induce any significant modification in the mean age of the vaginal opening (Figure 2A) as determined by one-way ANOVA (F = 2.748, p = 0.0541). With respect to the percentage of animals with vaginal opening (Figure 2B), the two-way ANOVA showed significant time (F = 38.48; p < 0.0001; df = 18) and treatment (F = 10.83; p < 0.0001; df = 3) effects. Based on Bonferroni post hoc test, 47 mg/kg BW tartrazine showed a significant difference (p < 0.05) in the percentage of animals with vaginal opening as compared with the Control group. This group displayed 100% of vaginal opening vs. 41.66% for control on day 45.

Lines 239 to 248

3.3. Relative weight of ovary and uterus

The main effects on reproductive organs are presented in Figure 3. The results of the relative uterine weight indicate that there is a statistically significant difference between groups as determined by one-way ANOVA (F = 6.098, p = 0.0057). Dunnett Post Hoc multiple comparisons test showed that the difference between tartrazine at the dose of 47 mg/kg BW (Figure 3B) and the control group is statistically significant (p < 0.05). Tartrazine increased significantly the relative uterine weight at the dose of 47 mg/kg BW (Figure 3B). Whatever, after 40 days of treatment, tartrazine had no significant effect on the relative weight of the ovaries (Figure 3A) as determined by one-way ANOVA (F = 2.793, p = 0.0739).

Lines 254 to 263

3.4. Epithelial thickness of the uterus and vaginas

The main reproductive effects are summarized in Figure 4. The results of the uterine epithelial thickness indicate that there is a statistically significant difference between groups as determined by one-way ANOVA (F = 6.602, p = 0.0041). Dunnett Post Hoc multiple comparisons test showed a significant difference between tartrazine at the dose of 27 (p < 0.05) and 47 mg/kg BW (p < 0.01) as compared to the control group (Figure 4A). This difference is also confirmed by the microphotographs presented in Figure 4C. However, one-way ANOVA indicated that sacrificed at the estrus phase, the administration of tartrazine had no significant effect (F = 0.6609, p = 0.5880) on the vaginal epithelial thickness (Figure 4B)

Lines 295 to 305

3.6. Ovarian follicles

The main effects on follicular growth are summarized in Table 1 and Figure 6. Table I and Figure 6 show the number of total follicles and different types of follicles, and the microphotographs of ovaries after 40 days of treatment with tartrazine respectively. One-way ANOVA indicated a significant difference between groups. The difference was reflected on the number of total follicles (F = 8.831, p = 0.0011), primary follicles (F = 9.771, p = 0.0007), secondary follicles (F = 4.744, p = 0.0149), and antral follicles (F = 4.329, p = 0.0205). Dunnett Post Hoc multiple comparisons test showed a significant increase in the total number of follicles (p < 0.01), primary follicles (p < 0.01), secondary follicles (p < 0.01), and antral follicles (p < 0.05) with tartrazine at a higher dose (47 mg/kg BW) as compared to the control group.

Lines 328 to 332

3.7. Ovarian total cholesterol and proteins

Figure 7 represents the ovarian total cholesterol and protein after 40 days of treatment. One way ANOVA indicated that ovarian total cholesterol (F = 0.6152, p = 0.6151) and protein (F = 0.6086, p = 0.6190) were not significantly affected following treatments (Figure 7).

Lines 337 to 346

3.8. Hormone levels

The main effects on hormone serum concentrations are summarized in Figure 8. One-way ANOVA indicated a significant difference between groups. The difference was reflected on the LH serum concentration (F = 10.85, p = 0.0004), and estradiol serum concentration (F = 130.1, p < 0.0001). Dunnett Post Hoc multiple comparisons test showed a significant increase in LH (p < 0.001) (Figure 8B) and Estradiol (p <  0.001) (Figure 8C) serum concentration at the dose of 47 mg / kg BW as compared to the control group. However, FSH (F = 2.619, p = 0.0866) and Progesterone (F = 0.2903, p = 0.8318) serum concentrations were not significantly affected by the treatment with tartrazine at all tested doses (Figures 8A and 8D).

b) At each beginning of each result, it begins with some phrases as shown for example in L212-213, 221-225, etc... but the statistical analysis should not be repeated in results, therefore that should be in Materials and Methods in statistical analysis. Furthermore, I repeat again, that the main effects and the other fixed effects and interactions were not presented.

The statistical analysis has been removed at the beginning of each result. The main effect, the fixed effect, and the interactions have been added to the manuscript. Attached is the answer in the new manuscript:

Line 203 to 211

2.7. Statistical analysis

Data were expressed as mean ± standard error on the mean (SEM). A two-way ANOVA repeated measures followed by Bonferroni post-hoc tests was used to compare the effect of tartrazine on body weight and the percentage of animals with vaginal opening. The fixed effects or factors were treatment (each individual dose of tartrazine vs. control group), time or periods of analysis, and their interaction. ANOVA one-way followed by Dunnet’s test (when appropriate) was used for the other data with treatment as a fixed effect. All of these tests were performed using GraphPad Prism 5.03 software (La Jolla, CA, USA, 2009). Differences were considered significant at p ˂ 0.05.

Lines 213 to 220

3.1. Bodyweight of animals

Figure 1 shows the effect of tartrazine exposure on body weight evolution throughout the period of treatment. Two-way ANOVA indicated a significant time effect (F = 108.3; p < 0.0001; df = 13), a non-significant treatment effect (F = 0.2327; p = 0.8736; df = 3) and a non-significant interaction effect (F=0.1026; p > 0.9999; df=39). In addition, Bonferroni correction multiple comparison test indicated that all doses of tartrazine were neither effective (p > 0.9999) in increasing, nor reducing body weight suggesting that the significant time effect observed is due to the normal growth of animals over time.

Lines 225 to 234

3.2. Vaginal opening

The main endocrine effect on the onset of puberty is summarized in Figure 2. Compared to the control group, the treatment did not induce any significant modification in the mean age of the vaginal opening (Figure 2A) as determined by one-way ANOVA (F = 2.748, p = 0.0541). With respect to the percentage of animals with vaginal opening (Figure 2B), the two-way ANOVA showed significant time (F = 38.48; p < 0.0001; df = 18) and treatment (F = 10.83; p < 0.0001; df = 3) effects. Based on Bonferroni post hoc test, 47 mg/kg BW tartrazine showed a significant difference (p < 0.05) in the percentage of animals with vaginal opening as compared with the Control group. This group displayed 100% of vaginal opening vs. 41.66% for control on day 45.

Lines 239 to 248

3.3. Relative weight of ovary and uterus

The main effects on reproductive organs are presented in Figure 3. The results of the relative uterine weight indicate that there is a statistically significant difference between groups as determined by one-way ANOVA (F = 6.098, p = 0.0057). Dunnett Post Hoc multiple comparisons test showed that the difference between tartrazine at the dose of 47 mg/kg BW (Figure 3B) and the control group is statistically significant (p < 0.05). Tartrazine increased significantly the relative uterine weight at the dose of 47 mg/kg BW (Figure 3B). Whatever, after 40 days of treatment, tartrazine had no significant effect on the relative weight of the ovaries (Figure 3A) as determined by one-way ANOVA (F = 2.793, p = 0.0739).

Lines 254 to 263

3.4. Epithelial thickness of the uterus and vaginas

The main reproductive effects are summarized in Figure 4. The results of the uterine epithelial thickness indicate that there is a statistically significant difference between groups as determined by one-way ANOVA (F = 6.602, p = 0.0041). Dunnett Post Hoc multiple comparisons test showed a significant difference between tartrazine at the dose of 27 (p < 0.05) and 47 mg/kg BW (p < 0.01) as compared to the control group (Figure 4A). This difference is also confirmed by the microphotographs presented in Figure 4C. However, one-way ANOVA indicated that sacrificed at the estrus phase, the administration of tartrazine had no significant effect (F = 0.6609, p = 0.5880) on the vaginal epithelial thickness (Figure 4B)

Lines 295 to 305

3.6. Ovarian follicles

The main effects on follicular growth are summarized in Table 1 and Figure 6. Table I and Figure 6 show the number of total follicles and different types of follicles, and the microphotographs of ovaries after 40 days of treatment with tartrazine respectively. One-way ANOVA indicated a significant difference between groups. The difference was reflected on the number of total follicles (F = 8.831, p = 0.0011), primary follicles (F = 9.771, p = 0.0007), secondary follicles (F = 4.744, p = 0.0149), and antral follicles (F = 4.329, p = 0.0205). Dunnett Post Hoc multiple comparisons test showed a significant increase in the total number of follicles (p < 0.01), primary follicles (p < 0.01), secondary follicles (p < 0.01), and antral follicles (p < 0.05) with tartrazine at a higher dose (47 mg/kg BW) as compared to the control group.

Lines 328 to 332

3.7. Ovarian total cholesterol and proteins

Figure 7 represents the ovarian total cholesterol and protein after 40 days of treatment. One way ANOVA indicated that ovarian total cholesterol (F = 0.6152, p = 0.6151) and protein (F = 0.6086, p = 0.6190) were not significantly affected following treatments (Figure 7).

Lines 337 to 346

3.8. Hormone levels

The main effects on hormone serum concentrations are summarized in Figure 8. One-way ANOVA indicated a significant difference between groups. The difference was reflected on the LH serum concentration (F = 10.85, p = 0.0004), and estradiol serum concentration (F = 130.1, p < 0.0001). Dunnett Post Hoc multiple comparisons test showed a significant increase in LH (p < 0.001) (Figure 8B) and Estradiol (p <  0.001) (Figure 8C) serum concentration at the dose of 47 mg / kg BW as compared to the control group. However, FSH (F = 2.619, p = 0.0866) and Progesterone (F = 0.2903, p = 0.8318) serum concentrations were not significantly affected by the treatment with tartrazine at all tested doses (Figures 8A and 8D).
